# Regional intergenerational mobility and corporate innovation: Evidence from China

**Changfu Luo🆔\*, Lian Xie**

Economics and Management School, Wuhan University, Wuhan, China

\* luochfu@whu.edu.cn

**Data Availability Statement:** The data supporting the findings of this study are owned by the Center for Social Survey (Sun Yat-sen University). Data are available through cssdata@mail.sysu.edu.cn with permission of the Center for Social Survey. In

## Abstract

By stimulating social vitality and improving innovation efficiency, intergenerational mobility plays an essential role in economic development. With the data from China Labor-force Dynamic Survey (CLDS) and A-share listed companies, this paper uses the methodology of intergenerational order correlation to measure regional intergenerational mobility, and examines the impact of regional intergenerational mobility on corporate innovation. The results are as follows: (1) Regional intergenerational mobility promotes corporate innovation both quantitatively and qualitatively, and a series of robustness tests confirm our findings; (2) Two channels identified are government-enterprise human capital allocation and fairness perception; (3) Heterogeneity analysis shows that the innovation effect of regional intergenerational mobility is more significant in the high-tech industry and private enterprises; (4) Multi-dimensional market-oriented policy can be used to get rid of the shackles of low intergenerational mobility on corporate innovation. Our findings provide implications for developing countries on how to address the relationship between inequality and economic development.

## Introduction

Since the reform and opening-up, China's economic and social development has made remarkable achievements. However, with the continuous growth of wealth, social problems, such as inequality, have become increasingly prominent [1]. According to data released by the World Bank, income inequality accelerated globally between 1980 and 2018, and the Gini coefficient of China, an emerging economy, has always been higher than the international warning line. Existing research has shown that higher inequality is strongly associated with lower intergenerational mobility, which may hinder class transition by enhancing intergenerational inheritance [2]. Therefore, in order to break through the current stratification and intergenerational solidification, it is urgent to create a healthy and fair development environment.

Currently, economic growth is increasingly dependent on innovation [3, 4]. [5] suggest that technological innovation contributes to a higher level of economic growth with labor productivity improvements. Enterprises are the main innovation participants and play an irreplaceable role in the national economy. As proposed by [6], technological innovation is a process in which individuals integrate their human capital with enterprise resources. The excess profit-sharing mechanism provides incentives for management and employees to commit themselves

addition, the firm-level data used in our study come from Chinese Research Data Service. Readers can apply for it from http://www.cnrds.com with the permission of Chinese Research Data Service.

**Funding:** The author(s) received no specific funding for this work.

**Competing interests:** The authors have declared that no competing interests exist.

to innovative activities [7, 8]. For enterprises, innovation is an internal strategic choice under the comprehensive action of external factors [9, 10]. Through technological innovation, enterprises can improve their core competitiveness and win the market competition. Therefore, a fair development environment is a valid driver of innovation [11–13].

Intergenerational mobility represents the correlation between family background and offspring income, and is considered to measure environmental justice [14, 15]. On the one hand, intergenerational mobility regulates the allocation of human capital between regions and departments by affecting individuals' employment choices and migration preferences [16]. On the other hand, individuals' fairness perceptions are closely related to intergenerational mobility [17]. Specifically, areas with higher intergenerational mobility enable local residents to have more opportunities for upward mobility. Vulnerable groups can prevent the intergenerational transmission of poverty through their own efforts instead of family background. In this case, the individual's sense of fairness is also enhanced. Conversely, lower intergenerational mobility makes it difficult for people to change their fortunes, even if they can achieve the class transition. As a result, generational inequality has worsened. People's employment choices and migration preferences are adjusted accordingly, resulting in inter-regional and inter-departmental human capital flows. From the perspective of enterprises, talent supply in the external labor market is restricted by regional intergenerational mobility. In addition, intergenerational mobility may directly affect management's innovation decisions and employees' innovation efforts within enterprises. Therefore, reasonable intergenerational mobility closely associated with fairness can provide a favorable environment for corporate development, and ultimately contribute to economic growth through corporate innovative activities.

Corporate innovation is a powerful driving force for economic development, so what role does regional intergenerational mobility play? Moreover, some scholars have found significant differences in intergenerational mobility among American states. To improve the accuracy of our research, we need to pay attention to intergenerational mobility at the city level. Focusing on this issue, we investigate the impact of regional intergenerational mobility on corporate innovation and analyze the potential channels. This paper may contribute to the literature in three ways. First, the existing studies on corporate innovation mainly concentrate on macroeconomic fluctuations, industrial policy, and culture. Starting with intergenerational mobility at the city level, we are committed to refining the research on the corporate innovation environment and integrating the research framework in this field from the macro-level to the micro-level. Second, prior literature has studied chiefly the measurement of indicators and explored which factors affect intergenerational mobility. This paper demonstrates the role of regional intergenerational mobility in promoting corporate innovation, which expands the previous literature. Finally, we also investigate the internal mechanism of regional intergenerational mobility affecting innovation output from supply and demand for corporate innovation. Our conclusions contribute to deepening the understanding that equity of opportunity can stimulate the innovation vitality of enterprises and talents, providing a policy reference for the government to smooth the channels for talent development.

The remainder of this paper is organized as follows. Section 2 conducts theoretical analysis and proposes hypotheses. Section 3 describes the data and measurement method for regional intergenerational mobility. Section 4 investigates the impact of regional intergenerational mobility on corporate innovation and analyzes possible mechanisms. Section 5 concludes this paper and provides policy implications.

## Theoretical analysis and hypotheses formalization

### Regional intergenerational mobility and corporate innovation

Enterprises' innovative willingness and decisions are influenced by a variety of factors, among which internal factors include executive characteristics [18], firm size [19], and ownership structure [20]. Besides, external factors such as industrial policy [21], macroeconomic fluctuations [22], and the market environment [23] are also considered to be necessary. These factors portray the development environment in which enterprises' innovative activities are located from different perspectives. Enterprise resources are divided into tangible resources (e.g., raw materials, machinery and equipment) and intangible resources (e.g., reputation, environment). From the perspective of resource dependence [24], demonstrated that the environment of an enterprise is an important indicator of comparative advantage. Without the financial and environmental resources required for successful innovation activities, it is difficult for enterprises to achieve a high level of innovation output [25, 26]. In addition, stakeholder theory holds that corporate activities are closely connected with and even influenced by various stakeholders (including employees, customers and society) [27, 28]. Therefore, a fair development environment enables market mechanisms to play a decisive role in resource allocation and stimulate enterprises' innovative vitality by giving full play to the functions of various market and non-market resources [29].

Regional intergenerational mobility represents the opportunity for local people to realize upward mobility and reflects the openness of the local social class structure. Higher intergenerational mobility provides everyone more opportunities to compete fairly, and its impact permeates the innovation activities of enterprises from individual behavioral decisions. Specifically, the impact of regional intergenerational mobility on corporate innovation can be summarized as follows:

First, people in areas with high intergenerational mobility are more likely to achieve class transition through their efforts instead of family background. When the belief that hard work can succeed is deeply rooted in the hearts, people usually put in more effort. Especially among high-quality groups, their behavioral decisions are often susceptible to the fairness of the city where they live [30]. Therefore, regions with high intergenerational mobility attract many high-quality groups, including scientific talents, by conveying the information that they possess in a reasonably competitive environment [31]. From the perspective of the enterprise, more high-quality managers and employees are provided. Moreover, the open social class structure also implies that the innovation efforts of enterprise managers and employees can be effectively stimulated, which helps to improve enterprise innovation output further.

Second, in order to obtain fairer opportunities for development, enterprises and individuals are generally willing to migrate to areas with higher intergenerational mobility. However, whether it can be translated into actual action depends on the constraints. For enterprises, relocation risks, difficulties and operational losses usually lead to caution in final decision-making [32, 33]. For individuals, China's household registration segmentation and urban settlement thresholds create barriers to relocation [34]. Under this scenario, many enterprises and individuals, although their early development is limited to local areas, are more willing to focus on technological innovation to lay the economic and technological foundation for the later migration to religions with high intergenerational mobility. Accordingly, we put forward the following hypothesis:

**H1**: Regional intergenerational mobility positively affects corporate innovation. Namely, high intergenerational mobility has an incentive effect on innovative output.

As mentioned above, the active participation of enterprises and scientific research talents in cities with low intergenerational mobility can be interpreted as the preparation and effort to migrate to more equitable cities, which highlights the innovation effect of high intergenerational mobility. Furthermore, intergenerational mobility in a closed society influences local firm innovation even without considering inter-regional migration. We analyze from the talent supply side and the innovation demand side.

## Talent supply side: Human capital allocation

Intergenerational mobility is inherently associated with economic and social development, and moderate intergenerational mobility is positively significant to the accumulation and allocation of human capital. On the one hand, educational equity has attracted the attention of many scholars and governments [35, 36]. With human capital proven to be the engine of growth, open social structures mitigate the adverse effects of financial resource constraints on human capital accumulation, promoting economic growth and social development. On the other hand, economic interests are the primary factor influencing labor migration, which requires a fair social mobility mechanism to achieve and guarantee [37]. It means that equity of opportunity can trigger labor migration between regions, sectors, and occupations, However, the flow between firms and institutions is the most prevalent.

There is an optimal allocation ratio of human capital between the public and private sector. If entering the public sector becomes the first choice for many talents, the ratio of human capital allocation between two sectors will deviate, causing a series of economic consequences [38]. The career of social elites is highly sensitive to environmental justice, while low fairness easily leads to talent mismatch. It is worth noting that innovation efficiency is highly correlated with the level of human capital, while the productive private sector is a critical player in engaging in innovative mobility [39, 40]. Lower intergenerational mobility usually induces a massive influx of talent into the public sector, squeezing out the private sector, which directly weakens the innovation potential of enterprises. Thus, we posited:

**H2**: The allocation of human capital between institutions and enterprises is a channel through which regional intergenerational mobility influences corporate innovation.

## Innovation demand side: Fairness perception

Studies in behavioral economics have shown that individuals innately prefer fairness over rationality [41]. Moreover, assuming that fairness is a fundamental constraint in government policy decisions, a sense of unfairness arises when people struggle to reasonably be rewarded for their efforts. Equity theory believes that if employees perceive unfairness, especially in the R&D department, staffing efforts are likely to be reduced, which is not conductive to improving output efficiency [42]. Even if the enterprise invests more resources in R&D (e.g., capital and technical personnel), the negative work attitude caused by the sense of injustice may undermine the efficiency of investment. Therefore, we put forward the following hypothesis:

**H3**: Fairness perception is another channel through which regional intergenerational mobility influences corporate innovation.

## Research design

### Data source

This paper selected Shanghai and Shenzhen A-share listed companies as the research samples. We remove the sample companies of Special Treatment (ST) and Particular Transfer (PT) and

exclude companies in the financial industry that are barely innovative. The data used are derived from the following four sources: (1) We calculate regional intergenerational mobility based on data from China Labor-force Dynamic Survey (CLDS) in 2012, 2014 and 2016, which covers 29 provinces and direct-controlled municipalities in China, providing high-quality raw data for this study. (2) The data on patent applications and grants of A-share listed companies are sourced from the Chinese Research Data Services (CNRDS) Platform. (3) Firm-level data are collected from CSMAR databases. We use city codes representing A-share listed companies are registered to match intergenerational mobility and the CNRDS database. In order to eliminate the perturbation of outliers, we winsorize the relevant variables at 1%.

## Variable measurement

**Corporate innovation.** Previous studies have used R&D expenditures as a proxy variable for corporate innovation, while the number of patent applications can effectively measure enterprise innovation ability [43]. The reasons are listed as follows: First, some scholars have found that many A-share listed companies still have insufficient disclosure of R&D investment [44]. Second, corporate innovation is generally high input and high uncertainty, indicating that R&D expenditures is are challenging to convert innovation output effectively, resulting in measurement errors.

According to the rules of Chinese patent law, invention patents belong to the high level of innovation and are generally regarded as a proxy variable of innovation quality. We simultaneously apply the total number of patent applications to represent the amount of enterprise innovation. Referring to [45], we constructed the following variables based on three types of patents: (1) *Patent_total* represents the total number of patent applications for three types. (2) *Patent_fm* represents invention patent count, and it is generally used to measure the core innovation ability of enterprises. (3) *Patent_ffm* represents the sum of utility-model patent counts and external-design patent counts, indicating corporate innovation ability with relatively low requirements. After adding 1 respectively, the above three variables are logarithmic in the empirical analysis. Similarly, we construct three proxy variables based on patent granted counts in the robustness test, including *Patent1_total*, *Patent1_fm*, and *Patent1_ffm*.

**Regional intergenerational mobility.** Measuring intergenerational mobility accurately has been a critical issue in related research, and the methods proposed in the previous literature are summarized as follows: First, Intergenerational income elasticity is used as a proxy variable for intergenerational mobility, reflecting the level of equal opportunity [46, 47]. However, considerable literature is skeptical of this. [48] have suggested that estimating intergenerational income elasticity places higher requirements on sample data, because the lack of long-term tracking data may cause measurement errors. Many scholars also argue that measuring intergenerational income elasticity requires income data within an individual's life cycle, while data limitations force us to replace permanent income with current income, resulting in life cycle bias [49]. Then, to avoid attenuation bias, some literature measures intergenerational mobility in terms of education, occupation, and health and attempts to argue for the importance of intergenerational mobility [50]. Lastly, recent studies have begun to measure social mobility from directed hierarchical mobility [51]. Intergenerational mobility is usually divided into relative mobility and absolute mobility. Absolute mobility refers to the expected achievement of children given their parents' income levels. By contrast, relative mobility focuses on the relative differences between children from different family backgrounds. However, it is important to point out that the increase in relative mobility is probably result from the fact that outcomes for children from low-income families get worse. In such circumstances, higher relative mobility is not necessarily desirable. We, thus, use absolute mobility as a proxy variable

for regional intergenerational mobility. In order to measure intergenerational mobility and compare it regionally, following [52], this paper constructs intergenerational mobility indicators at the city level, and the measurement process is as follows.

First, using the social status of parents and children to estimate intergenerational order correlations between regions. Based on the fact that subjective social status can more accurately describe social class, we merge the China Labor-force Dynamic Survey (CLDS) data in 2012 and 2014 from the city's perspective. After excluding tracking samples and those that have migrated, we selected individual's subjective social status in the questionnaire to replace the income variable. Eq (1) is given by:

$$R_{ic} = \alpha_c + \beta_c P_{ic} + \varepsilon_{ic} \tag{1}$$

Where $R_{ic}$ and $P_{ic}$ denotes the social status ranking of children and parents in city $c$ within the same group, respectively. Besides, $\beta_c$ represents the slope of the rank-rank relationship, measuring the degree of relative mobility in city $c$. $\alpha_c$ is the intercept term.

Then, since the rank-rank relationship is linear [53], we exploit $\alpha_c$ and $\beta_c$ in Eq (1) to calculate the absolute mobility of children at $p$ percentile in city $c$. Thus, Eq (2) can be expressed as follows:

$$r_{pc} = \alpha_c + \beta_c p \tag{2}$$

Following the discussion above, we define absolute intergenerational mobility $r_{pc}$ as the expected rank of a child living in city $c$ with parents whose social status rank is in the p percentile. Higher absolute intergenerational mobility indicates that children have more space to move upwards, while the influence of family background becomes less important. In other words, if the absolute mobility of city $c$ higher, residents are more likely to have access to fair competition.

It is worth mentioning that among all social classes, upward mobility at the bottom of the distribution deserves special attention, which is a crucial starting point for the government to formulate policies. Following [53], we calculate absolute mobility at the 25[th] percentile to characterize regional intergenerational mobility.

Table 1 provides descriptive statistics characteristics of the main variables. It can be seen that the average logarithmic of patent applications, representing innovation output, is 1.2615. The comparison of the minimum and maximum values implies significant differences in enterprises' innovation abilities. Absolute intergenerational mobility ranges from 7.0824 to 23.3473, reflecting the apparent differences between cities.

**Control variables.** Considering the potential influence of other factors on corporate innovation at the firm and city level, we carefully select several firm-level and city-level controls. The firm-level controls include: (1) rate of return on total assets (*ROA*), asset-liability ratio (*Loar*) and Tobin's Q ratio (*Q*), which is used to control the impact of corporate performance and growth; (2) number of employees (*Labor*) and age of the enterprise to go public (*IPO_year*) are considered proxy variables for firm size; (3) R&D expenditures (*R&D*) and the shareholding ratio of the largest shareholder (*Stockholder*) are used separately to control innovation input and ownership structure. The city-level controls include: (1) the level of economic development (*GDP*), expressed as the logarithm of GDP; (2) industrial structure, measured by the ratio of the tertiary sector to gross output. Besides, we also control the fixed effect of years and industries.

Table 1 reports the descriptive statistical characteristics of the main variables. The mean value of listed companies' innovation output (*Patent_total*) is 1.3307, but the minimum and maximum values are 0 and 7.4759, respectively. In addition, we also find that the intergenerational mobility levels vary significantly among cities.

**Table 1. Summary statistics.**

| Variable | Mean | Sd | Min | Max |
|---|---|---|---|---|
| Panel A: Firm level | | | | |
| Patent_total | 1.2651 | 1.5266 | 0 | 8.6785 |
| Patent_fm | 0.8575 | 1.2266 | 0 | 8.5824 |
| Patent_ffm | 0.8992 | 1.3161 | 0 | 7.8026 |
| ROA | 0.0679 | 0.0666 | -0.1945 | 0.2681 |
| Loar | 0.4321 | 1.8808 | 0 | 142.7178 |
| Q | 1.9854 | 1.2084 | 0.9172 | 8.8037 |
| Labor | 7.2817 | 1.3172 | 0.6931 | 12.4380 |
| IPO_year | 1.6521 | 1.1675 | 0 | 3.3673 |
| R&D | 17.3768 | 1.3932 | 6.8352 | 25.0252 |
| Stockholder | 3.4848 | 0.4632 | 2.1751 | 4.3238 |
| Panel B: City level | | | | |
| $Abmobility_{25}$ | 13.2362 | 2.5334 | 7.0824 | 23.3473 |
| GDP | 17.8692 | 1.0907 | 12.8608 | 19.6049 |
| Tertiary_ratio | 0.5079 | 0.0896 | 0.100 | 0.8098 |

## Benchmark model

To investigate the impact of regional intergenerational mobility on corporate innovation, we construct the following *Tobit* model:

$$Innovation_{i,j,c,t} = \alpha_0 + \beta_1 Abmobility_{c,p} + \sum \beta_m X_{i,j,c,t,m} + \sigma_t + \sigma_j + \varepsilon_{i,j,c,t} \qquad (3)$$

Where $i$ is the enterprise, $j$ is the industry, $c$ is the city, and $t$ is the year. $Innovation_{i,j,c,t}$ represents the logarithmic number of enterprises' patent applications. $Abmobility_{c,p}$ is the independent variable, denoting that the absolute mobility of city $c$ at the $p$ percentile. $X_{i,j,c,t,m}$ is a series of control variables. $\sigma_t$ and $\sigma_j$ are the year fixed effect and industry fixed effect, respectively. $\varepsilon_{i,j,c,t}$ is the error term.

## Empirical results and analyzes

**Baseline results.** Using Eq (3), we examine the impact of regional intergenerational mobility on corporate innovation. As shown in Table 2, column (1) reports the quantitative effect without control variables. Absolute mobility is significantly positively correlated with corporate innovation at the level of 1%, indicating that higher intergenerational mobility boosts local firms to produce more patents. The estimated results are still robust when we add firm-level variables and city-level variables in column (2). In addition, the influence of regional intergenerational mobility on enterprises' innovative ability is also investigated, and columns (3)-(4) showed regression results. As we can see, the estimated coefficients of absolute mobility are 0.0431 and 0.0391, respectively, and significant at 1%, which implies that a fairer competitive environment can significantly improve the innovation level of local enterprises.

In summary, we believe that regional intergenerational mobility contributes to both the quantity and quality of innovation in firms, providing empirical evidence to support H1.

## Mechanism analysis

This study has shown that higher intergenerational mobility benefits local firm innovation and theoretically analyzed its impact mechanisms. In this section, we demonstrate two channels of human capital allocation and fairness perception through empirical analysis, providing a more

**Table 2. Results of baseline regression.**

| Dependent Variable | Patent_total | | Patent_fm | Patent_ffm |
|---|---|---|---|---|
| | (1) | (2) | (3) | (4) |
| Abmobility$_{25}$ | 0.0413** (0.0120) | 0.0375*** (0.0121) | 0.0431*** (0.0119) | 0.0391*** (0.0130) |
| ROA | | -0.0255 (0.3427) | 0.3902 (0.3432) | 0.1624 (0.3806) |
| Loar | | -0.3154** (0.1553) | -0.1621 (0.1536) | -0.2545 (0.1068) |
| Q | | 0.0388*** (0.0133) | 0.0470*** (0.0128) | 0.0392*** (0.0149) |
| R&D | | 0.3390*** (0.0231) | 0.3859*** (0.0235) | 0.2297*** (0.0254) |
| Labor | | 0.1689*** (0.0234) | 0.1599*** (0.0307) | 0.2262*** (0.0343) |
| IPO_year | | -0.3495*** (0.0263) | -0.3575*** (0.0260) | -0.2989*** (0.0289) |
| Stockholder | | 0.0083*** (0.0017) | -0.0028* (0.0016) | 0.0143*** (0.0018) |
| GDP | | 0.1222*** (0.0341) | 0.1517*** (0.0339) | 0.0589 (0.0373) |
| Tertiary_ratio | | 0.4282 (0.3569) | 1.2371*** (0.3541) | -0.6197 (0.3865) |
| Year FE & Industry FE | Yes | Yes | Yes | Yes |
| _cons | -7.2594*** (0.4365) | -9.5124*** (0.6806) | -11.3693*** (0.6789) | -7.7332*** (0.7422) |
| Obs | 7809 | 7809 | 7809 | 7809 |
| Pseudo R$^2$ | 0.1066 | 0.1099 | 0.1106 | 0.1282 |

Note: (1) ***, **, * indicates the significance of 0.01, 0.05, and 0.1, respectively; (2) Robust standard errors in parentheses.

comprehensive explanation of how regional intergenerational mobility affects corporate innovation.

**Human capital allocation.** Referring to the method proposed by [1], we use data from China Household Finance Survey (CHFS) in 2015 to calculate the average education level of government and enterprise employees in the sample cities. In this way, the proxy variable for the inter-departmental human capital allocation ratio can be obtained. Specifically, individuals working in government agencies, organizations or public institutions are defined as working in public departments. In contrast, individuals working in collective firms, privately or individually-owned businesses, and private firms are defined as working in private department. Besides, we assign values to educational levels respectively, as detailed in Appendix A in S1 File Correspondingly, we can obtain an indicator of the ratio of human capital allocation between governments and firms in the sample cities.

The results are shown in Table 3. First, the estimated results at the city level in column (1) indicate that intergenerational mobility significantly increases the proportion of local workers entering the private sector. To be precise, higher intergenerational mobility brings more upward mobility opportunities, which helps attract more highly educated talents into the private sector and provides intellectual support for innovative activities. After adding explanatory and mediating variables in the model simultaneously, we get the results displayed in columns (2)-(4). It can be seen that corporate innovation is positively shaped by intergenerational mobility and the ratio of government-firm human capital allocation. Besides, the estimated coefficients of regional intergenerational mobility are much smaller than the one we get from baseline results in Table 2. Hence, we can infer that improving the local government-firm human capital allocation ratio is a channel for intergenerational mobility to promote corporate technological innovation. H2 is generally supported.

**Fairness perception.** Based on the theoretical analysis in Section 2, another potential channel behind our findings is fairness perception. Likewise, we exploit micro-data to estimate metrics of fairness perception in the city in which the enterprise is located. With reference to [54], we assign values sequentially based on five options in the 2015 China Household Finance

**Table 3. Mediating effect tests: Human capital allocation.**

| Dependent Variable | Human capital allocation | Innovation | | |
|---|---|---|---|---|
| | Hcapital | Patent_total | Patent_fm | Patent_ffm |
| | (1) | (2) | (3) | (4) |
| Abmobility$_{25}$ | 0.0054*** (0.0005) | 0.0288*** (0.0123) | 0.0380*** (0.0122) | 0.0285** (0.0246) |
| Hcapital | | 1.5955*** (0.4435) | 0.9758** (0.4434) | 1.9605*** (0.4845) |
| Firm-level control variables | No | Yes | Yes | Yes |
| City-level control variables | Yes | Yes | Yes | Yes |
| Industry FE | No | Yes | Yes | Yes |
| Year FE | No | Yes | Yes | Yes |
| Obs | 7809 | 7809 | 7809 | 7809 |
| Pseudo R$^2$ | 0.7624 | 0.1104 | 0.1108 | 0.1289 |

Note: (1) ***, **, * indicates the significance of 0.01, 0.05, and 0.1, respectively; (2) Robust standard errors in parentheses.

Survey (CHFS) on how individuals perceive social equity, where "very unfair" equals 5 and "very fair" equals 1, obtaining a proxy variable for fairness perception at the city level. A lower value of this proxy variable denotes that the city is fairer on average.

The results are shown in Table 4. In column (1), the impact of intergenerational mobility at the city level on fairness perception is examined, indicating that individuals perceive a fairer level in areas of higher intergenerational mobility. Higher intergenerational mobility makes local people more likely to achieve the class transition, and for this reason, they believe in fair opportunities to motivate their own efforts instead of family background. Similarly, for enterprises, a higher evaluation of social fairness promotes more motivation for corporate management and R&D talents to carry out technological innovation and thus obtain more innovative benefits. Replace the dependent variable with patent application counts, and the results are shown in columns (2)-(4). As we can see, when our model includes mediating variable *Justice*, the coefficients of intergenerational mobility are positive and significant at the level of 1%. Combined with the baseline results in Table 2, we calculate that the indirect effects in columns (2)-(4) are 0.316, 0.231, and 0.306, respectively. Our findings confirm that fairness perception is another important channel by which regional intergenerational mobility influences corporate innovation. Therefore, H3 is also supported.

**Table 4. Mediating effect tests: Fairness perception.**

| Dependent Variable | Fairness perception | Innovation | | |
|---|---|---|---|---|
| | Justice | Patent_total | Patent_fm | Patent_ffm |
| | (1) | (2) | (3) | (4) |
| Abmobility$_{25}$ | -0.0207*** (0.0010) | 0.0554*** (0.0127) | 0.0565*** (0.0126) | 0.0570*** (0.0138) |
| Justice | | 0.8461*** (0.1947) | 0.6317*** (0.1927) | 0.8428*** (0.2125) |
| Firm-level control variables | No | Yes | Yes | Yes |
| City-level control variables | Yes | Yes | Yes | Yes |
| Industry FE | No | Yes | Yes | Yes |
| Year FE | Yes | Yes | Yes | Yes |
| Obs | 7809 | 7809 | 7809 | 7809 |
| Pseudo R$^2$ | 0.4850 | 0.1106 | 0.1110 | 0.1289 |

Note: (1) ***, **, * indicates the significance of 0.01, 0.05, and 0.1, respectively; (2) Robust standard errors in parentheses.

## Robustness checks

Due to endogenous problems such as omitted variables and reverse causation in empirical research, biased results are caused. Thus, we conduct a series of robustness checks to refine our research design.

**Omitted variables.** Given that this study uses micro-data to measure intergenerational mobility at the city level and the active and large-scale labor migration between cities, especially in China's first-tier cities, measurement errors are likely to arise. However, the social class structure of cities may be closely related to local culture and ideology. Numerous studies have confirmed that culture shapes innovation, so ignoring cultural influences may lead to endogenous problems [55, 56]. Therefore, we make the following attempts: First, according to the net inflow and outflow data of various provinces and cities in the China Migrants Dynamic Survey (CMDS) from 2011 to 2016, the top 5 cities in terms of net population inflow and the top 3 provinces in terms of migration rate are excluded to eliminate potential interference in population migration. Throughout all the columns in Table 5, the estimated coefficients of regional intergenerational mobility are significantly positive, implying that our findings remain robust.

Second, we select performance-oriented and future-oriented culture indices at the province level taken from [57] as proxy variables for regional culture and add them to the model to control the influence of culture. Appendix B in S1 File shows the details of cultural indicators at the province level. The reasons are stated as follows: (1) GLOBE defines performance orientation and future orientation as the extent to which society encourages members to pursue gains and futures; (2) these cultural indicators are theoretically closely related to innovative motivation. Columns (1)-(3) in Table 6 include future-oriented culture indicators, while columns (4)-(6) represent performance-oriented culture indicators in the model. The results show a significant positive correlation between regional intergenerational mobility and corporate innovation, further indicating that benchmark results are robust.

**Instrumental variable test.** Enterprises continue to increase R&D investment and actively carry out innovative activities, which enables enterprises to receive economic innovation benefits and enhances the innovation ability of their cities, leading to rapid economic development [58]. Notably, economic growth is closely linked to changes in social structure, suggesting that corporate innovation may also affect intergenerational mobility in regions [59], and endogeneity concerns arise. Referring to [54], this study chooses a dummy variable as an instrumental variable to counter endogeneity. The dummy variable is defined as whether the city, where the enterprise located in, is a commercial port that was forced to open between 1842 and 1922. The reasons are as follows: (1) the Qing government was forced to open many

**Table 5. Results of controlling population migration.**

| Dependent Variable | Excluding the top five cities in terms of net population inflow | | | Excluding the top three provinces by population migration rate | | |
|---|---|---|---|---|---|---|
| | Patent_total | Patent_fm | Patent_ffm | Patent_total | Patent_fm | Patent_ffm |
| | (1) | (2) | (3) | (4) | (5) | (6) |
| Abmobility$_{25}$ | 0.0291** (0.0132) | 0.0297** (0.0130) | 0.0404*** (0.0143) | 0.0358*** (0.0138) | 0.0296** (0.0134) | 0.0541*** (0.0148) |
| Firm-level and city-level control variables | Yes | Yes | Yes | Yes | Yes | Yes |
| Industry FE & Year FE | Yes | Yes | Yes | Yes | Yes | Yes |
| Obs | 5054 | 5054 | 5054 | 4377 | 4377 | 4377 |
| Pseudo R$^2$ | 0.1215 | 0.1249 | 0.1393 | 0.1253 | 0.1314 | 01429 |

Note: (1) ***, **, * indicates the significance of 0.01, 0.05, and 0.1, respectively; (2) Robust standard errors in parentheses.

**Table 6. Results of controlling regional culture.**

| Dependent Variable | Patent_total | Patent_fm | Patent_ffm | Patent_total | Patent_fm | Patent_ffm |
|---|---|---|---|---|---|---|
| | (1) | (2) | (3) | (4) | (5) | (6) |
| Abmobility$_{25}$ | 0.0512*** (0.0129) | 0.0481** (0.0128) | 0.0544*** (0.0138) | 0.0476*** (0.0123) | 0.0502** (0.0122) | 0.0481*** (0.0132) |
| Wldx | 1.2862*** (0.4875) | 0.5779*** (0.4909) | 1.6157*** (0.5288) | | | |
| Jxdx | | | | 1.0610*** (0.2873) | 0.9865*** (0.2842) | 1.0895*** (0.3176) |
| Firm-level and city-level control variables | Yes | Yes | Yes | Yes | Yes | Yes |
| Industry FE & Year FE | Yes | Yes | Yes | Yes | Yes | Yes |
| Obs | 7500 | 7500 | 7500 | 7500 | 7500 | 7500 |
| Pseudo R$^2$ | 0.1105 | 0.1109 | 0.1272 | 0.1108 | 0.1114 | 0.1273 |

Note: (1) ***, **, * indicates the significance of 0.01, 0.05, and 0.1, respectively; (2) Robust standard errors in parentheses; (3) Wldx, Jxdx represent future-oriented and performance-oriented culture indicators at the province level, respectively.

cities into commercial ports, and foreign politics, economy, and culture strongly impacted these closed cities, driving changes in the local social structure. Because regional social structure has historical continuity, the instrumental variable *FOP* satisfies the relevance assumption. (2) Whether a city used to be a treaty port does not affect the current business decisions of local enterprises, and the instrumental variable *FOP* thus satisfies the exogeneity assumption.

Table 7 reports the results of 2SLS regressions using *FOP* as an instrumental variable. As we can see, the first-stage regression results in column (1) show that *FOP* is significantly negatively correlated with regional intergenerational mobility. The results in columns (2)-(4) indicate that regional intergenerational mobility significantly increases the innovation output of local firms. Therefore, our conclusions are still supported.

**Other robustness checks.** To ensure the robustness of the baseline results, we further conduct the following tests: First, the listed companies may have false patent applications, that is, the number of the patent granted is less than that of applications. In this paper, three dependent variables, *Patent1_total*, *Patent1_fm* and *Patent1_ffm*, are reproduced using the patent grant count instead of the application count. Second, [52] proposed that the intergenerational mobility ranking between cities has strong stability, and it is difficult to fluctuate sharply in a short period. Nonetheless, the intergenerational mobility we measured using micro-data from 2012 to 2014 does not explain well the output of corporate innovation before 2012. To this end, this study limits the observational sample of listed companies to after 2011.

**Table 7. Instrumental variable test.**

| Dependent Variable | First Stage | Second Stage | | |
|---|---|---|---|---|
| | Abmobility$_{25}$ | Patent_total | Patent_fm | Patent_ffm |
| | (1) | (2) | (3) | (4) |
| FOP | -0.4816*** (0.0779) | | | |
| Abmobility$_{25}$ | | 0.8007*** (0.2114) | 0.8593** (0.2159) | 0.7718*** (0.2205) |
| Firm-level control variables | Yes | Yes | Yes | Yes |
| City-level control variables | Yes | Yes | Yes | Yes |
| Industry FE | Yes | Yes | Yes | Yes |
| Year FE | Yes | Yes | Yes | Yes |
| Obs | 7809 | 7809 | 7809 | 7809 |
| F-Test | 70.60 | | | |

Note: (1) ***, **, * indicates the significance of 0.01, 0.05, and 0.1, respectively; (2) Robust standard errors in parentheses.

**Table 8. Ther robustness tests.**

| Dependent Variable | Replacing the dependent variable | | | Excluding samples before 2011 | | |
|---|---|---|---|---|---|---|
| | Patent1_total | Patent1_fm | Patent1_ffm | Patent_total | Patent_fm | Patent_ffm |
| | (1) | (2) | (3) | (4) | (5) | (6) |
| Abmobility$_{25}$ | 0.0384*** (0.0113) | 0.0388*** (0.0114) | 0.0600*** (0.0130) | 0.0385*** (0.0126) | 0.0319** (0.0130) | 0.0535*** (0.0134) |
| Firm-level and city-level control variables | Yes | Yes | Yes | Yes | Yes | Yes |
| Industry FE & Year FE | Yes | Yes | Yes | Yes | Yes | Yes |
| Obs | 7809 | 7809 | 7809 | 5852 | 5852 | 5852 |
| Pseudo R$^2$ | 0.1218 | 0.1086 | 0.1294 | 0.1099 | 0.1345 | 0.1331 |

Note: (1) ***, **, * indicates the significance of 0.01, 0.05, and 0.1, respectively; (2) Robust standard errors in parentheses.

As shown in Table 8, regional intergenerational mobility is significantly positively correlated with corporate innovation, which implies that higher intergenerational mobility effectively contributes to the innovation output of local firms quantitatively and qualitatively. These results confirm our findings.

## Heterogeneity tests

To better understand whether the impact of intergenerational mobility on corporate innovation varies due to corporate heterogeneity. In this section, we conduct group tests based on differences in the industry to which the enterprise belongs and corporate ownership. Furthermore, market-based reforms are considered effective in improving inequality, so we also discussed the impact of reforms with different biases on the relationship between intergenerational mobility and corporate innovation.

**Industrial heterogeneity.** The inherent nature of the high-tech industry requires companies to equip themselves with more substantial innovative capabilities and win the market competition with continuous technological innovation. However, the city's social environment in which the enterprise is located cannot be ignored [60]. From the perspective of enterprises, higher intergenerational mobility makes enterprise management have a strong desire and motivation to innovate, and employees are more actively involved in innovative practices. Relatively speaking, due to the low requirements and dependence of traditional industries on technological innovation, the innovative enthusiasm of enterprises is generally not high, so the impact of the local social environment on patent output is limited. To verify the above analysis, we construct a dummy variable *Hightech* to characterize industrial differences (high-tech industry equals 1) based on the 75th percentile of the enterprises' patent application count. Accordingly, the interaction term of *Abmobility$_{25}$* × *Hightech* is introduced into the model, and the results are shown in Table 8. As can be seen from columns (1)-(3), the estimated coefficients of the interaction terms are all significantly positive at 1%, indicating that regional intergenerational mobility promotes corporate innovation more prominently in the high-tech industry (Table 9).

**Firm ownership.** In China, state-owned enterprises (SOEs) and private enterprises (PEs) are the two pillars of economic development. However, there are apparent differences between the two types of enterprises regarding resource access, institutional logic and interest protection [61, 62]. It can be summarized as follows: First, state-owned enterprises have inherent advantages in obtaining talent resources, credit resources, and market resources, resulting in less pressure when facing technological innovation competition. In contrast, resource disadvantages force private enterprises to improve their innovative capability by resorting to the positive externalities of higher social mobility. Second, since state-owned enterprises are

**Table 9. Heterogeneity test of industry.**

| Dependent Variable | Patent_total | Patent_fm | Patent_ffm |
|---|---|---|---|
| | (1) | (2) | (3) |
| Abmobility$_{25}$ × Hightech | 0.0316*** (0.0117) | 0.0643*** (0.0130) | 0.0366*** (0.0135) |
| Abmobility$_{25}$ | -0.0281*** (0.0090) | -0.0347*** (0.0104) | -0.0329*** (0.0108) |
| Hightech | 2.3590*** (0.1578) | 1.5481*** (0.1750) | 2.1755*** (0.1825) |
| Firm-level and city-level control variables | Yes | Yes | Yes |
| Industry FE & Year FE | Yes | Yes | Yes |
| Obs | 7809 | 7809 | 7809 |
| Pseudo R$^2$ | 0.2901 | 0.2608 | 0.2880 |

Note: (1) ***, **, * indicates the significance of 0.01, 0.05, and 0.1, respectively; (2) Robust standard errors in parentheses.

generally government-led, their innovative activities may primarily respond to policy orientation, limiting the external environment. However, higher intergenerational brings more possibilities to the public. The market-oriented institutional logic drives private enterprises to participate in technological innovation activities and improve their core competitiveness, so as to obtain rich, innovative benefits. Finally, private enterprises do not provide employees with stable jobs and enough welfare benefits compared with state-owned enterprises. Both PEs and their employees are more motivated to innovate to achieve economic gains. Based on the above analysis, we infer that the incentive effect of regional intergenerational mobility on corporate innovation may be weakened among state-owned enterprises. In this regard, the interaction term of regional intergenerational mobility and firm ownership (private enterprises, SOE equal to 1) is introduced into the model. The results of columns (1)-(3) in Table 10 show that the coefficients for the interaction terms are significantly positive. Therefore, we believe that the innovative effect of intergenerational mobility on local private enterprises is higher than that of state-owned enterprises.

**Market-oriented reform.** Generally, there are many policy directions for market-oriented reforms, but their focus is different. For example, some cities are committed to developing non-state-owned economies, aiming to fully tap market potential by improving product quality, which is common in eastern China. On the contrary, China's coastal areas are more open, and it is a wise choice to innovate in quantity quickly but not necessarily in quality, so these cities may pay more attention to the development of product markets. In addition, much of the literature suggests that market and social factors play different roles in economic development. Since an appropriate and effective market-oriented can enhance the economic growth

**Table 10. Heterogeneity test of firm ownership.**

| Dependent Variable | Patent_total | Patent_fm | Patent_ffm |
|---|---|---|---|
| | (1) | (2) | (3) |
| Abmobility$_{25}$ × SOE | 0.0316*** (0.0117) | 0.0643*** (0.0130) | 0.0366*** (0.0135) |
| Abmobility$_{25}$ | -0.0281*** (0.0090) | -0.0347*** (0.0104) | -0.0329*** (0.0108) |
| SOE | 2.3590*** (0.1578) | 1.5481*** (0.1750) | 2.1755*** (0.1825) |
| Firm-level and city-level control variables | Yes | Yes | Yes |
| Industry FE & Year FE | Yes | Yes | Yes |
| Obs | 7809 | 7809 | 7809 |
| Pseudo R$^2$ | 0.2901 | 0.2608 | 0.2880 |

Note: (1) ***, **, * indicates the significance of 0.01, 0.05, and 0.1, respectively; (2) Robust standard errors in parentheses.

Table 11. Heterogeneity test of market-oriented reform.

| Dependent Variable | Patent_total | | Patent_fm | | Patent_ffm | |
|---|---|---|---|---|---|---|
| | (1) | (2) | (3) | (4) | (5) | (6) |
| $Abmobility_{25} \times Market1$ | -0.0283** (0.0129) | | -0.0288** (0.0130) | | -0.0169 (0.0139) | |
| $Abmobility_{25} \times Market2$ | | -0.0309* (0.0162) | | -0.0065 (0.0161) | | -0.0568*** (0.0174) |
| $Abmobility_{25}$ | 0.3100** (0.1248) | 0.3053** (0.1418) | 0.3204*** (0.1252) | 0.0964 (0.1405) | 0.2018 (0.1345) | 0.5356*** (0.1526) |
| Market1 | 0.4381** (0.1888) | | 0.4328** (0.1894) | | 0.2606 (0.2033) | |
| Market2 | | 0.4127* (0.2372) | | -0.0020 (0.2352) | | 0.8445*** (0.2557) |
| Firm-level and city-level control variables | Yes | Yes | Yes | Yes | Yes | Yes |
| Industry FE & Year FE | Yes | Yes | Yes | Yes | Yes | Yes |
| Obs | 7809 | 7809 | 7809 | 7809 | 7809 | 7809 |
| Pseudo $R^2$ | 0.1102 | 0.1102 | 0.1108 | 0.1108 | 0.1283 | 0.1287 |

Note: (1) ***, **, * indicates the significance of 0.01, 0.05, and 0.1, respectively; (2) Robust standard errors in parentheses.

effect of social factors and make up for the shortcomings of lag in social structure to a certain extent, the role between them may be alternative or complementary [63]. Specific to this paper, is the relationship between regional intergenerational mobility and corporate innovation affected by market-oriented reforms with different biases? To answer this question, we introduce the following two interaction terms in the model, namely $Abmobility_{25} \times Market1$ and $Abmobility_{25} \times Market2$. Among them, Market1 and Market2, taken from the Chinese Research Data Services (CNRDS) Platform, represent the index of market-oriented reform at the province level that is biased towards the development of non-state-owned enterprises and product markets respectively. See Appendix C in S1 File for details.

In Table 11, the results of columns (1)—(2) show that both Market1 and Market2 are positively correlated with Patent_total, while the coefficients of the interaction terms are significantly negative. As expected, when low social mobility is not conducive to corporate innovation, marketization can play a role in alternative governance and make up for social shortcomings. However, market-oriented reforms with different biases generally have differentiated innovation effects, so we speculate that alternative governance mechanisms may also be heterogeneous. Accordingly, this study subdivides patents into invention and non-invention, examining our conjectures from the perspective of patent heterogeneity. In comparison, we find that the coefficients of the interaction terms are only significantly negative in column (3) and column (6). Our interpretation is that market-oriented reforms tend to support the development of non-state-owned economies. Its alternative governance mechanism breaks down structural barriers to social mobility by guiding enterprises to engage in substantial innovations and enhance social vitality. In an environment of low social mobility, market-oriented reforms biased toward product markets may induce enterprises to achieve rapid innovation in quantity, which may not be conductive to improving the overall innovative capacity.

## Conclusion and policy implications

Moderate social mobility can make a society vibrant and economically resilient. In this paper, intergenerational mobility at the city level is measured using data from China Labor-force Dynamic Survey (CLDS) to empirically investigate the impact of regional intergenerational mobility on corporate innovation. The conclusions are as follows: (1) Regional intergenerational mobility has an incentive effect on corporate innovation, which significantly increases the innovative output of local firms in both quantity and quality. Several robustness tests confirm our findings. (2) Mechanism analysis shows that regional intergenerational mobility

mainly affects corporate innovation through two channels: government-enterprise human capital allocation and fairness perception. (3) Taking into account heterogeneity, we find that the role of high generational mobility in promoting corporate innovation is more pronounced in high-tech industries and private enterprises. (4) Market-oriented reform and regional intergenerational mobility have an alternative relationship in stimulating corporate innovation, but there is obvious heterogeneity.

The implications of our conclusions can be summarized below. First of all, in order to cope with inequality in developing countries, especially in China, continuing to promote the reform of the social system and smooth the upward mobility channel can stimulate the vitality of social innovation and achieve sustainable economic development. Secondly, China's demographic dividend is gradually fading, posing a serious challenge to economic growth. Thus, we should create a healthy and fair developmental environment, prevent structural mismatch of talents, and further release talent dividends. Finally, promoting the rational and orderly flow of talents is not achieved overnight, and alternative governance mechanisms for market factors should be recognized. Through multi-dimensional market-oriented policy tools to stimulate enterprises' innovative potential, we can make up for the shortcomings of imbalance in social structure.

There are some limitations to this study. Firstly, referring to the existing literature, we try to measure the regional intergenerational mobility. Due to the limitation of data acquisition, only cross-sectional proxy variables can be obtained in this paper. Although there is a lot of evidence that intergenerational mobility at the city level is relatively stable in the short term, we cannot perfectly solve the doubts caused by non-temporal variables. Secondly, the research sample of this study is limited to Chinese listed companies that need to be strictly regulated. Among industrial enterprises above designated size, the proportion of listed companies is relatively small. If future studies extend the sample to all industrial enterprises above size, other findings may be possible. Finally, we also can compare whether the impact of regional intergenerational mobility on corporate innovation is different between OECD countries and emerging countries.

## Supporting information

**S1 File. Appendix A.** Classification of academic qualifications. **Appendix B.** Cultural indicators of each province. **Appendix C.** Market_oriented reform indicators of each province in 2014.
(DOCX)

## Author Contributions

**Conceptualization:** Changfu Luo.

**Data curation:** Changfu Luo, Lian Xie.

**Formal analysis:** Changfu Luo, Lian Xie.

**Methodology:** Changfu Luo.

**Software:** Changfu Luo.

**Supervision:** Changfu Luo, Lian Xie.

**Writing – original draft:** Changfu Luo.

**Writing – review & editing:** Changfu Luo, Lian Xie.

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
