## [Decision Letter · Decision Letter 0]

13 Feb 2023

PONE-D-22-31829Regional intergenerational mobility and corporate innovation: Evidence from ChinaPLOS ONE

Dear Dr. Luo,

Thank you for submitting your manuscript to PLOS ONE. After careful consideration, we feel that it has merit but does not fully meet PLOS ONE’s publication criteria as it currently stands. Therefore, we invite you to submit a revised version of the manuscript that addresses the points raised during the review process.

We look forward to receiving your revised manuscript.

Kind regards,

Bing Xue, Ph.D.

Academic Editor

PLOS ONE

Journal Requirements:

This study was supported by the National Social Science Foundation of China (20ARK006).

Reviewers' comments:

Reviewer's Responses to Questions

**Comments to the Author**

1. Is the manuscript technically sound, and do the data support the conclusions?

Reviewer #1: Yes

Reviewer #2: Yes

Reviewer #3: Yes

2. Has the statistical analysis been performed appropriately and rigorously? 

Reviewer #1: Yes

Reviewer #2: Yes

Reviewer #3: Yes

3. Have the authors made all data underlying the findings in their manuscript fully available?

Reviewer #1: Yes

Reviewer #2: Yes

Reviewer #3: Yes

4. Is the manuscript presented in an intelligible fashion and written in standard English?

Reviewer #1: Yes

Reviewer #2: Yes

Reviewer #3: Yes

5. Review Comments to the Author

Reviewer #1: This paper examines the impact of regional intergenerational mobility on corporate innovation based on the data from China Labor-force Dynamic Survey (CLDS) and A-share listed companies. They find that regional intergenerational mobility promotes corporate innovation both quantitatively and qualitatively, and government-enterprise human capital allocation and fairness perception are the two channels. I recommend that the paper can be accepted by the journal after minor revision. Before the publication, several errors should be corrected by the authors.

1. The introduction needs to be adjusted. Specifically, lines 66-77 in the original text need to be adjusted to the end of the text. Generally speaking, in the introduction, you only need to introduce the background and highlight the innovation of the article, without presenting the research results.

2. The research literature review is slightly weak and needs to be updated to 2022. More support is given in the literature review section.

3. “firm”, "Company" and “enterprise” need more accurate expression, or unified expression.

4. In the part of Empirical Results and analyzes, it is suggested to directly use variable meanings rather than abbreviations, which is more convenient for readers to understand. For example: Abmobility25, Hcapital, etc.

5. Set the tables according to journal guidelines.

6. The authors are advised to add the limitations of this study and possible future research directions after the conclusions and policy implications.

Reviewer #2: Dear authors,

Based on the title of the manuscript, it looks promising and can be an insightful study. However, there are some problems to be further improved as well. Please consider the following points to improve your work.

(1) In the abstract section, I note that regional intergenerational mobility has a positive impact on both quantity and quality of firm innovation. However, the authors do not explain this in section3.2.1. Detailed information is required.

(2) In Section 2.1, the authors also discuss the possible negative relationship between high regional intergenerational mobility and firm innovation. I think the hypothesis of H1B is not reliable, and it is not convenient for readers to understand the paper clearly. Therefore, I suggest this hypothesis can be removed.

(3) In Section 2.3, I suggest that the authors should be more detailed in their theoretical analysis.

(4) In Section 3.2.2, the authors should explain why absolute intergenerational mobility is chosen as the proxy variable for regional intergenerational mobility.

(5) In lines 281-282, for the sentence “Besides, we also controls the fixed effect of years and industries.”, the pronoun “we” must be used with a non-third-person form of a verb “control”.

(6) In line 309, it should be “As we can see”.

(7) In line 332, there is a spelling mistake in the word “obtanined”.

Reviewer #3: This paper investigates the impact and mechanism of regional intergenerational mobility on corporate innovation. Higher regional intergenerational mobility is found to have a significant positive impact on the innovative activities of local firms. This paper proposes the possible mechanisms for this finding, such as human capital allocation and fairness perception. This is an interesting topic and I have the following comments.

The authors need to add a sentence explaining the proposed methodology. Keywords must be written in alphabetical order.

The authors stated that this study follows a prior study that measures regional intergenerational mobility at the city level. However, they need to explain why this study should measure regional intergenerational mobility at the city level.

Significance (Table 2 and following) is not a %, but a number: 0.01, 0.05 etc.

The authors should add the study limitations and suggestions for future studies.

6. PLOS authors have the option to publish the peer review history of their article (what does this mean?). If published, this will include your full peer review and any attached files.

Reviewer #1: No

Reviewer #2: No

Reviewer #3: No

---

## [Author Response · Author response to Decision Letter 0]

5 Mar 2023

Dear Editors and Reviewers,

First of all, please allow us to take this opportunity to express our heartfelt thanks to you for taking time out from your busy schedule to review this manuscript. You have provided us with constructive comments and suggestions, which are of great help for us to further improve this manuscript. We have carefully reviewed and revised the manuscript according to your valuable comments and suggestions. Here, we explain the revised work in detail below and provide the point-by-point responses to the reviewers’ comments. These changes have been provided in the file labeled "Response to Reviewers".

---

## [Editor Report · Decision Letter 1]

13 Mar 2023

Regional intergenerational mobility and corporate innovation: Evidence from China

PONE-D-22-31829R1

Dear Dr. Luo,

We’re pleased to inform you that your manuscript has been judged scientifically suitable for publication and will be formally accepted for publication once it meets all outstanding technical requirements.

Kind regards,

Bing Xue, Ph.D.

Academic Editor

PLOS ONE
---

## [Editor Report · Acceptance letter]

23 Mar 2023

PONE-D-22-31829R1 

Regional intergenerational mobility and corporate innovation: Evidence from China 

Dear Dr. Luo:

I'm pleased to inform you that your manuscript has been deemed suitable for publication in PLOS ONE. Congratulations! Your manuscript is now with our production department. 

Kind regards, 

on behalf of

Professor Bing Xue 

Academic Editor

PLOS ONE